# Towards a Characterization of Background Music Audibility in Broadcasted TV

**DOI:** 10.3390/ijerph20010123

**Published:** 2022-12-22

**Authors:** Roser Batlle-Roca, Perfecto Herrera-Boyer, Blai Meléndez-Catalán, Emilio Molina, Xavier Serra

**Affiliations:** 1Music Technology Group, Universitat Pompeu Fabra, 08002 Barcelona, Spain; 2BMAT Music Innovators (BMAT Licensing, S.L.), 08028 Barcelona, Spain

**Keywords:** background music, loudness perception, psychoacoustic experiments, complex auditory scene, everyday life environments, broadcasted TV, listening conditions, behaviour and cognition

## Abstract

In audiovisual contexts, different conventions determine the level at which background music is mixed into the final program, and sometimes, the mix renders the music to be practically or totally inaudible. From a perceptual point of view, the audibility of music is subject to auditory masking by other aural stimuli such as voice or additional sounds (e.g., applause, laughter, horns), and is also influenced by the visual content that accompanies the soundtrack, and by attentional and motivational factors. This situation is relevant to the music industry because, according to some copyright regulations, the non-audible background music must not generate any distribution rights, and the marginally audible background music must generate half of the standard value of audible music. In this study, we conduct two psychoacoustic experiments to identify several factors that influence background music perception, and their contribution to its variable audibility. Our experiments are based on auditory detection and chronometric tasks involving keyboard interactions with original TV content. From the collected data, we estimated a sound-to-music ratio range to define the audibility threshold limits of the *barely audible* class. In addition, results show that perception is affected by loudness level, listening condition, music sensitivity, and type of television content.

## 1. Introduction

Music broadcasted in audiovisual productions provides a large stream of income for the music industry through author copyrights. The distribution rules of royalties vary by country and consider several conditions, such as time slots and the importance of music in production.

In 2017, WIPO (World Intellectual Property Organization) determined that non-audible or inaudible background music should not generate distribution rights, and *barely audible* background music should generate 50% of the standard value generated by audible music [1]. Therefore, the assessment of background music audibility in broadcasted TV is under scrutiny since copyright remuneration is compromised on the audibility level. Indeed, collective management organisations, e.g., SGAE (General Society of Authors and Publishers of Spain), still need to detect when these circumstances are present in broadcasted programs.

### 1.1. Problem Identification

The music industry faces the problem of proposing a definition for the three audibility categories: *audible*, *barely audible* and *inaudible* background music, and requires automatic systems that are able to classify the perceived audibility of background music in audiovisual productions.

From the perspective of the audience, the received signal is a mixture emitted by TV broadcasters. That signal contains both musical and non-musical sources simultaneously, despite the music flows being separated from the other sounds at the moment of the production (Figure 1). Hence, it is difficult to measure the perceived loudness level of each signal source independently without applying source separation, or considering technical characteristics such as masking and frequency selectivity that strongly influence what the end user hears.

Moreover, one might think that the definitions of the categories *audible*, *barely audible* and *inaudible* background music only depend on their relative loudness thresholds. That is, determining which loudness level or loudness range of the music must be classified into each category. If music could be separated from the rest of the audio content and a simple sound-to-music ratio (SMR) could be obtained through loudness estimation of each track, we could classify the clips according to their corresponding audibility level, after establishing a loudness threshold between these categories.

However, other influential elements, such as environmental noise, listening conditions, and even the musical sensitivity of the listeners, might increase or decrease this required threshold of loudness. Hence, we face a complex auditory scene analysis problem with many variable elements that can influence music perception.

### 1.2. Background Concepts and State-of-the-Art

Understanding background music perception is complex and affected by several characteristics, which must be considered in this study. Therefore, we must be careful in defining which of these elements influence music perception and how these may condition the audibility level of music in audiovisual content. Thus, this subsection’s objective is to lay out the principal elements of perception that are taken into account here, along with similar investigations that support our hypotheses and the experiments’ approaches addressed in Section 2.

In broadcasted TV programs, it is common to face auditory masking, as there are many circumstances where there are multiple sounds simultaneously transmitted, e.g., a presenter introducing a contest while there is music in the background and the public is applauding, which might overwhelm the audience with multiple sounds. To distinguish them, they rely on frequency selectivity, which refers to the ability to separate frequency components of a complex sound [2]. Nonetheless, this does not assure they are perceiving all the sounds in the mixture, as one might mask the others.

In addition, loudness is the perceived magnitude of a sound and cannot be directly measured since it is a subjective quantity [2]. Thus, the loudness perception of a signal is subjective, and each subject’s perception can vary due to individual features such as attention and awareness (with attention referring to voluntarily focusing the mind on some stimulus or feature of it (e.g., music pitch, following a melody), whereas awareness refers to being conscious that something, such as background music, is happening or is present).

Regardless, thanks to the auditory scene analysis [3], the human auditory system can *stream* or tell apart different sonic objects even when they are concurrently stimulating our hearing sense (e.g., the speech of a TV conductor, laughs from a live audience in the TV set, some background music in the program, and the telephone ringing at home) [4]. According to our current knowledge, the complex sonic stimulation is usually simplified into foreground and background (and sometimes an extra additional stream can be extracted too) by means of automatic or semi-automatic heuristics based on hypotheses about structural properties of concurrent sounds (i.e., harmonicity, synchronicity, spatial placement, etc.) and how they evolve over time (i.e., if they covariate or not, if they have different spectral content, etc.). Once streams are created, attentional processes might produce further refinements on one of them, or might even suppress the processing of part of the available information.

Moreover, a known phenomenon is the so-called “inattentional unawareness” [5], consisting of misperceiving unexpected stimuli, due to a lack of attention or sensorial resources. In the auditory domain, we find the idea of *inattentional deafness*. As exposed by Koreimann [5], inattentional deafness in music is defined as the inability to consciously perceive characteristics of the music piece, for example, unawareness of a given instrument when attention is directed towards a different one, or to decide on rhythmic features. Their investigation sustains the existence of inattentional deafness in music and that the human auditory system is more influenced by attentional distraction than it was previously thought. Additionally, Molloy et al. [6] describes inattentional deafness as the failure to perceive auditory stimuli under high visual load. Indeed, several studies carried out by Lavie and colleagues ([6,7,8,9]) demonstrate the influence of visual stimuli on inattentional deafness through different behavioural experiments.

Hence, for our exposed scenario (loudness perception of background music in broadcasted TV programs), these investigations justify the need to study different perceptual characteristics, such as motivated attention and perceptual unawareness, and the influence of the visual stimuli on our definition of the *barely audible* loudness thresholds. However, this last element (i.e., visual stimuli) is herein omitted since it is beyond the scope of the study and not of priority in our research agenda.

Another element to consider in this investigation is ecological validity. In empirical evaluations, ecological validity is understood as the amount of realism an experiment setup has, compared to the person’s real context [10]. It relies on the similarities between the testing conditions and the real scenario where the studied element occurs. Consequently, when conducting auditory and behavioural experiments, it is important to preserve the ecological validity of the case study to be able to extrapolate and generalise the obtained results.

During this investigation, we considered the most pragmatic context of our case study: watching broadcasted TV content while seated on a sofa viewed on a television. Nonetheless, we consider other listening scenarios that are also frequent when engaging in audiovisual stimulation at home: loudspeakers and headphones, as people also frequently watch TV programs from computers, tablets, or mobile phones.

### 1.3. Related Work

Background music audibility estimation is not a widely addressed topic, but during the development of this study, SGAE (which is the most important music rights management society in Spain) conducted a similar investigation with a research team from Universitat Politècnica de València (UPV). Their work has been reported in the 152nd Audio Engineering Society Convention held in May 2022 [11]. In this article, López and Ramallo expose the same industrial problem as us: studying the audibility of background music in television programs considering three levels of audibility (*audible*, *barely audible* and *inaudible*). They consider several elements that may influence the perception of background music and conclude that the feasible solution is to build a deterministic system that converges to the average estimate of listeners’ audibility.

To evaluate the audibility level of background music, they created a manual annotation of 88 artificial 10-second clips of video, composed of the voice of two popular Spanish news hosts (one male, one female) and added songs of different music genres. Participants had to determine *barely audible* volume for each combination through an attenuator (sliding fader). The authors establish the threshold between *barely audible* and *inaudible* around −30 dB and propose a linear regression to adjust the perceived loudness level. They conclude that, when working with original broadcasted content where voice and music are mixed, it is required to use source separation techniques to estimate music loudness level. Unfortunately, from our own experience and from what has been reported in state-of-the-art ([12,13]), these techniques are still far from providing excellent results that can be applied in this sensitive scenario.

Additionally, considering all the previously discussed factors involved in a typical TV-watching situation, we think that the problem is still far from being solved with such a deterministic approach, and our aim is to further investigate towards a more nuanced and multi-component predictive model. Therefore, the goal of this investigation is to assess background music audibility in audiovisual productions by defining an approximate loudness level threshold for the three established auditory categories and to study the influence of listening conditions on background music perception, taking into account non-music signals (i.e., speech, applauses, etc.), environment noise, auditory scene-setting and subject-related characteristics. In this study, we report on two psychoacoustic experiments, based on auditory detection and chronometric tasks involving computer keyboard interactions, with real TV content. We identify that perception is influenced by loudness level, auditory condition, environment characteristics, ambience noise, music sensitivity and type of television content; and, establish two loudness thresholds that separate the three different audibility categories: *audible*, *barely audible* and *inaudible*.

## 2. Materials and Methods

To understand the exposed complex auditory scene analysis problem and evaluate the characteristics that influence background music perception, we planned, designed and carried out two psychoacoustic listening experiments: (1) *Did you hear the music?* and (2) *Perceiving the music*. In each experiment, certain factors are studied: sound frequency selectivity and SMR variation, in the first; and, listening conditions impact (headphones, loudspeaker, TV), in the second. Participants were submitted to different aural conditions from different audiovisual segments and had to assess their auditory perception through simple tasks involving computer keyboard-based answers.

The experiments were conducted onsite, always in the same room, and directly supervised by an investigator. To recreate a kind-of “living room”, the room included a TV and a sofa from which the participants attended the experiment. For the second experiment, a table and chair were set to complete the headphones and loudspeakers’ listening conditions. The average ambient noise in the room was 35.8 dBA, which agrees with the participants’ experience during the experiments, as most of them considered the noise environment weak or inaudible. Stimuli for both experiments are 10-second clips of video from five Spanish TV channels (TV1, TV2, LaSexta, Cuatro and Antena 3), recorded between 28th June 2021 at 13:15 and 7th July 2021 at 10:15. These stimuli were extracted from BMAT’s database and cannot be openly shared as they are subject to confidentiality regulations.

The software tool that we used to build and control all these experiments is lab.js (https://lab.js.org/ (accessed on 31 October 2022)), which offers the possibility to both write code using a high-level JavaScript library and to build experiments using an online graphical interface. Experiments were run online using the Pavlovia platform (https://pavlovia.org/ (accessed on 31 October 2022)).

Furthermore, the Institutional Committee for Ethical Review of Projects (CIREP) at Universitat Pompeu Fabra reviewed and approved the experimental approach as it complies with the ethical principles in research involving humans and personal data protection regulations (guided by Regulation (EU) 2016/679 of the European Parliament and of the Council of 27 April 2016 on the protection of natural persons with regard to the processing of data and on the free movement of such data, and repealing Directive 95/46/EC (General Data Protection Regulation)).

### 2.1. Participants

Following CIREP guidelines, we looked for participants between the ages of 18 and 65, who were Spanish speakers and did not report serious hearing problems. We consider a valid Spanish speaker anyone with a C1 level, that is, anyone who can express themselves fluently in the oral and written domain and can understand what they hear and read, even if they are not familiar with a topic. They were recruited through email announcements, social media communications and printed posters distributed around the University campus. An economical compensation of 10 USD per experiment participation was offered, which was paid through the Testable platform (https://www.testable.org/ (accessed on 31 October 2022)).

Before starting the listening test, a short questionnaire asked a series of demographic questions. Apart from registering their age, gender, nationality and language level, participants had to define their music sensitivity level by selecting one of the following categories:Non-musician—I don’t know how to read or play music/I rarely spend time listening to music.Music enthusiasts—I don’t know how to read or play music, but I am a music enthusiast. Hence, I listen to music often and I go to live music concerts.Amateur musician—I know how to read and play music (I play, at least, one musical instrument) and I enjoy listening to music, as well as going to live music concerts. However, I did not pursue professional studies.Intermediate musician—I play, at least, one instrument and I have some music studies background. Although, I do not earn enough money to consider myself a professional musician.Professional musician—I consider myself a professional musician, I can live from music-related activities and/or I have pursued or am pursuing professional music studies and I am able to play different instruments.

Until this point, we have presented the methodology common to both experiments. In the upcoming subsections, we describe specific details of materials and methods applied in each experiment separately.

### 2.2. Experiment 1

Experiment 1 *Did you hear the music?* wants to estimate the threshold limits of the category *barely audible*. That is, to obtain a specific loudness dB level where humans differentiate music from inaudible to *barely audible*, and from *barely audible* to audible.

We hypothesise that every participant has an upper and a lower threshold of the *barely audible* category, and we can extrapolate an average general threshold level from that. Additionally, the individual threshold may be influenced by our controlled variable “music sensitivity”.

For this experiment, with the help of the source separation system demucs [13], we separate the background music (music signal) from the specific content of the clips (e.g., voice, claps) (non-music signal). As the achieved separation with demucs is not always perfect, only clips where music and speech+sounds are well separated are selected. Then, we keep non-music signals as in the mixture and apply a linear increase (from −50 dB to −15 dB) or decrease (from −15 dB to −50 dB) in volume progression to the music signals. The experiment has two sections: ascending and descending, which are introduced randomly to each subject.

The separated stimuli (sound-music) were normalised at −14 LUFS/1dB Peak, and the volume progressions were applied afterwards. A volume calibration procedure was included at the beginning of the experiment to ensure good audibility throughout it. The calibration consisted in listening to a reference sound, which was normalised at a lower level (−40 LUFS/1dB Peak), extracted from an original TV scene, and had to adjust the device volume to hear it very low. Participants received the instruction: “Listen to the reference sound. Adjust the volume to hear it very low, without understanding what is said in it. That is, you must have the feeling that something sounds very faintly, and without the possibility of understanding it”.

Participants were presented with a chronometric task, where they were asked to press the keyboard spacebar as soon as they started to hear music (in the ascending section) or when they stopped hearing it (in the descending section). If participants did not press the spacebar at any time it was assumed as a null response. Each section contains 24 clips of 10-second videos, of which four appeared twice in one block, and four others were repeated in both blocks with opposed progression. We can estimate the SMR at which background music becomes audible or inaudible according to the moment the key was pressed and the volume progression (35 dB range), for each clip. Thus, the SMR level is not controlled by the participant but by us. We expect to estimate a feasible *barely audible* threshold range from this experiment.

#### 2.2.1. Cleaning Data

Several criteria were pre-defined to indicate potentially bad-quality data and the need to discard any suspicious data:High percentages of nulls (not answered threshold) in any of the sections of the experiment;An inverse threshold range (that is, obtaining a higher threshold for the descending section than for the ascending section), as we consider it not to be the logically expected majority trend.

After screening the available data, additional cutoff criteria were added. Since, we considered some music detections could happen right at the end of the clip, we set the threshold detection at volume progression limit (i.e., −15 dB and −50 dB), plus one extra second of reaction time (+3.5 dB):Subjects’ responses define a too-high threshold in the ascending section (Ath > −11.5 dB).Subjects’ responses define a too-low threshold in the descending section (Dth < −53.5 dB).

#### 2.2.2. Participants

This experiment had 42 participants, of which 38 (male = 17, female = 21) were considered valid participants. Participants were Spanish native speakers between 18 and 50 years old (mean = 24.32, median = 21.5), who mainly set the TV volume at 24 (approximately 58 dBA) and spent about 13 min completing the task. This time does not include the introduction to the experiment nor the payment explanation at the end of the experiment, which added about 10 min. Thus, in general, participants’ spent around 25 min at the experiment site.

Moreover, the participants were positioned so that their ears were situated at an average distance of 2.06 m from the TV. Furthermore, 42.1% of the participants considered themselves to be music enthusiasts, 23.7% of them music amateurs and 34.2% were music professionals.

### 2.3. Experiment 2

Experiment (2) *Perceiving the music* wants to quantify the effects of listening conditions or devices (headphones, a TV and computer speakers/loudspeakers) on the perception and awareness of background music and, consequently, on the variability of the *barely audible* threshold.

We hypothesise that headphones yield an easier detection of perceived background music than the other conditions. Therefore, the *barely audible* threshold will be lower, and its range will be narrower in this condition. All the participants used the same devices (headphones, TV and loudspeakers) and experienced all three conditions (same task). Hence, this is a within-subjects experiment.

For this experiment, the stimuli were normalised at −14 LUFS/1 dB Peak and a volume calibration procedure was included at the beginning of each listening condition. The volume calibration instructions were the same as in Experiment 1.

The proposed task consists of a detection and forced-choice task, where a 10-second clip of video appears, and participants must press the spacebar as soon as they detect music in the clip. Thus, reaction time helps us draw threshold levels for each category and listening condition. In other words, when a participant perceives music right at the beginning, we can derive it is audible; if it takes more time, it is *barely audible*; and if there is no reaction, the music can be considered inaudible.

However, the same stimuli cannot be shown in each listening condition as we would expose participants to the same content three times and induce learning. Consequently, we have three stimuli groups (A-B-C), of 24 clips each, which appear randomly, one for each condition (which are randomly distributed). That is, there are nine testing conditions (three auditory conditions per three stimuli groups).

At the end of the experiment, we include a subjective questionnaire about the hearing conditions (with only three possible answers—the listening conditions):In which listening condition did you hear better?In which listening condition did you feel more comfortable with the task?

#### 2.3.1. Participants

In this experiment, we had 60 participants. However, five participants were discarded due to erratic behaviour and underrepresentation. Thus, we will observe the results of 55 participants (male = 25, female = 30), of which all of them, but 2, were Native-Spanish speakers between 18 and 50 years old (mean = 23.2, median = 21.0).

Almost half of the participants (45.5%) considered themselves music enthusiasts, 18.2% music amateurs, 18.2% intermediate musicians and 18.2% music professionals. Moreover, participants’ ears were situated at an average distance of 2.06 m from the TV and 68 cm from the computer speaker. Furthermore, at each condition, after the volume calibration, participants slightly adjusted it to a subjectively comfortable level (the range of variation was of 15 dB between listening conditions) (Table 1).

Participants needed around 16 min to complete the experiment and spent 5 min on average in each listening condition. This time does not include the introduction to the experiment nor the payment explanation at the end of the experiment, which added about 10 min. Thus, in general, participants’ spent around 30 min at the experiment site.

Considering their perceptions during the experiment, on the one side, most participants recognised completing the task better with headphones (72.7%). Instead, 10.9% of the participants agreed it was with loudspeakers and another 10.9% with the TV. Nonetheless, 5.5% of participants could not decide on the best condition.

On the other side, we asked about the perceived synchronicity between their keyboard response and the moment when the music was heard or disappeared. Most of them indicated they did not know when they answered (34.5%) while the others mostly responded that they pressed the spacebar after (32.7%) or at the exact moment (29.1%) they recognised the music.

#### 2.3.2. Criteria Audibility Classification

The classification of clips per category was done depending on the reaction time (RT) when pressing the spacebar. After an observation of classification variability to establish the threshold between the *audible* and *barely audible* categories, it was decided that if a participant detected music within the 2 first seconds of the clip, it should be classified as *audible*. Instead, if they needed more time to detect background music, it is classified as *barely audible*, and if they did not press the bar at all it is classified as inaudible.

Clicks after the 11th second are considered misclassifications and the clips are assigned to the inaudible category.

## 3. Results

### 3.1. Experiment 1

Experiment 1 aimed to estimate the thresholds of the *barely audible* category employing exposure to a progressively ascending or descending volume in the background music. The experiment was divided into two blocks: one for the ascending music level condition and the other for the descending music level. From the first, we should estimate the upper level of the category and, from the second, the lower level.

#### 3.1.1. Average Threshold Level per Block

To calculate the average threshold level of each block, we took all the SMR classifications and obtained the average threshold level. For the ascending block, the mean was at −23.29 dB, and the descending block was at −36.76 dB.

It is rational to obtain such separate averages as each block corresponds to two different tasks: detecting background music without previous information (ascending) and having previous information (descending).

A one-way ANOVA (Table A1) was performed to compare the effect of the average threshold level obtained in each block. Our analysis revealed that there was a statistically significant difference (F(1479, 101) = [1.645], *p* = 0.001), meaning that a higher volume is required to detect music when not having any previous information, and a lower volume when there is previous information. This discrepancy is what helps us define the *barely audible* category threshold limits.

#### 3.1.2. *Barely Audible* Category Threshold Estimation

To estimate the *barely audible* threshold range, we pursued the idea of observing the overall threshold distributions on each block (Figure 2), and set the upper and lower limits from the 75% percentile. With this approach, we obtained an upper limit at −18.79 dB, from the ascending block, and a lower limit at −42.06 dB, from the descending. However, we found this threshold range to be too wide and impractical to determine the *barely audible* class.

Consequently, we decided to adopt a more conservative approach and base the limits on the median, to reflect the distribution of SMRs. Therefore, we obtained a rounded threshold of −22.7 dB to −36.4 dB (range = 13.7 dB, centre = −29.6 dB).

#### 3.1.3. Influence of Music Sensitivity and Type of Video Content

To determine if there exists an influence of subjects’ music sensitivity on background music detection, we performed a two-way ANOVA analysis on the obtained threshold level per block and music sensitivity level. For more consistency, we grouped participants into two categories: non-professionals (music enthusiasts and amateur musicians) and professionals (intermediate musicians and professional musicians). In Appendix A, Table A2 reflects there is a significant difference between music sensitivity level and experiment block (F(1, 1) = [1732.516], *p* = 0.000). Therefore, there is a clear influence of the music professionality of the participants on the detected thresholds. The more exposed to the music one is, the lower the background music detection threshold is found (Figure 3).

Furthermore, the presented stimuli were classified into the following type of video categories: advertisement (9), cooking show (1), documentary (5), news (4), reality show (1), talk show (6), sports (2), culture (1), weather (1) and TV contest (10). For news videos or the ones with someone talking constantly, a third of the participants expressed it was harder for them to detect the music because they were distracted by the voice or it was complicated to distinguish the music from the voice. Table 2 displays the average thresholds obtained for each category, considering the experiment block. We performed a two-way ANOVA analysis to evaluate the effect of each category on the determined threshold level, per block. The test revealed that there is a statistically significant interaction between the type of content (F(1, 9) = [17.775], *p* = 0.000).

### 3.2. Experiment 2

Experiment 2 intended to investigate the potential influences of different listening conditions: headphones, loudspeakers and TV. Therefore, the same task was performed in each condition: pressing the spacebar as soon as the music was perceived.

#### 3.2.1. Listening Condition Influence

To prove there is an effect of listening conditions on the speed of the keyboard response (and hence on the audibility of the music), we performed a one-way ANOVA analysis (Table A3). As we expected, there is a significant difference between the listening conditions (F(2, 2413) = [14.399], *p* = 0.000). Nonetheless, Tukey’s HSD test for multiple comparisons (Table A4) found that the mean value of listening conditions was not significantly different between the TV and loudspeakers conditions (*p* = 0.219, 95% C.I. = [−99.656, 586.619]). Another relevant finding is that the smallest reaction time is obtained with headphones whereas the highest corresponds to TV (Figure 4). According to the distance between the TV and the subjects, the time required for the sound wave to travel from the source to the human ear (max. 6 ms) is not large enough to justify the overall discrepancy.

Moreover, taking into account that the order of the conditions were randomly presented to each participant, we conducted another two-way ANOVA analysis of the listening sequences in perspective of the auditory conditions (as there is a significant difference between them). No significant differences were found (F(2, 5) = [1.109], *p* = 0.351), which guarantees that we can work with all the data together.

#### 3.2.2. Reaction Time and Audibility Categorization

To estimate the classification of clips into a determined audibility category, we set the threshold between the *audible* and *barely audible* classes at 2 s of reaction time. Regarding the *inaudible* class, we established, (and instructed the participants accordingly) that no interaction with the keyboard meant no music perception. In Figure 5, we can observe that with headphones participants have a faster median RT. Instead, both loudspeakers and TV conditions show a higher RT. We can acknowledge this difference for the *barely audible* classification: close to 4 s in headphones and close to 5 s for the other two.

## 4. Discussion

The principal goal of this investigation was to explore and attempt to define the *barely audible* music class in the context of broadcasted TV. That is, to bring light to the grey area between perceptual audible and inaudible background music. Considering that loudness is a subjective sensation and cannot be measured directly but must be done through estimations, we proposed two psychoacoustic experiments to capture the essence of “what is *barely audible*”. The introduced experiments validate that music perception is influenced by multiple characteristics, three of them (SMR, musical training, and listening conditions) have been analyzed herein.

From the first experiment, we estimated that the *barely audible* category could be delimited by SMR values between −22.7 dB and −36.4 dB. Thus, it can be established that if a SMR is higher than −22.7 dB, background music will generally be considered *audible*, and if it is lower than −36.4 dB, it will be *inaudible*.

Nonetheless, perception does not depend solely on loudness level, and therefore we explored different elements that could alter it. We detected an influence of music sensitivity towards the awareness of background music, as music professionals presented a lower detection threshold (−23.9 dB to −37.7 dB) in comparison to non-professionals (−22.0 dB to −35.6 dB). Thus, the proposed threshold can be considered an optimistic estimation (and that for the whole population will be slightly higher).

During our analysis, we observed music perception is influenced by the type of audiovisual material the audience is processing. Therefore, it would be interesting to have a wider and balanced representation of different program categories to extract robust conclusions about perceptual differences due to the type of content, a consideration that was not properly planned for when preparing these experiments.

Moreover, from Experiment 2, we derive a clear effect on listening conditions, especially when not using headphones (lowest inaudible classification threshold). Most participants expressed that it was easier to detect music with headphones, which could be due to their better frequency response (compared to the transducers used in the other conditions, and also to the lack of room reverberation).

In summary, we have tackled the infrequently studied problem of the blurry and labile limit between audibility and inaudibility. Our contributions improve, complement, and expand the methodology and findings reported in the scarce existing literature on it [11]. Here, we have estimated a range of SMR values that could define the thresholds for a *barely audible* category. Our findings provide convergent validity to those reported in that paper (−30 dB), but bear more ecological validity as we have used real-world TV clips, tested different listening conditions (which cause the thresholds to shift), and have demonstrated the important effect of listening sensitivity or musical training of the subjects on such thresholds. Finally, we have proposed and effectively used alternative techniques to experimentally address the problem.

## 5. Conclusions

This investigation provides a starting point for untangling background music perception in broadcasted TV, focused on establishing orientative thresholds between the three audibility categories set by WIPO: *audible*, *barely audible* and *inaudible*. Our approach takes into account the ecological validity of the experimental tasks used to understand this complex auditory scene analysis problem and centres on exploring data-driven definitions of a threshold level for an a priori ill-defined *barely audible* class, in addition to determining some of the influential elements on the three audibility categories.

We reported two psychoacoustic experiments, based on exploring the factors influencing the perception of background music: signal loudness level and listening conditions (headphones, loudspeakers, and TV set). From the obtained results, we proposed a threshold range for the *barely audible* category considering the level differences between the non-musical sound and the music, and proved that it is subject to the influence of the listening conditions. Therefore, the proposed limit from the first experiment is only valid for the TV condition. Moreover, we observed an effect of the music sensitivity level when perceiving background music, as the more one is exposed to music, the lower the threshold that one can detect the music. We hope this research draws attention to the need to further study background music audibility estimation regarding broadcasted TV.

## Figures and Tables

**Figure 1 ijerph-20-00123-f001:**
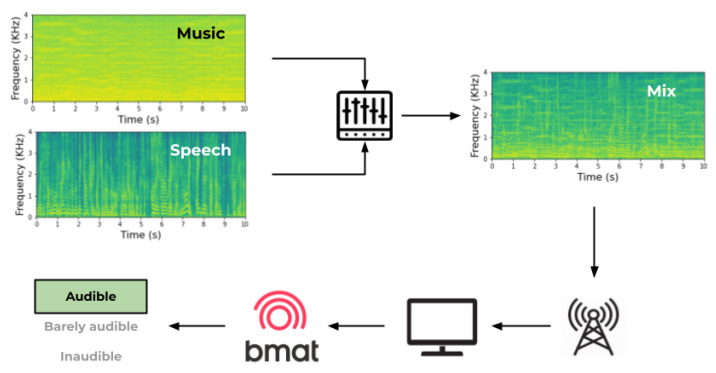
Representation of case scheme.

**Figure 2 ijerph-20-00123-f002:**
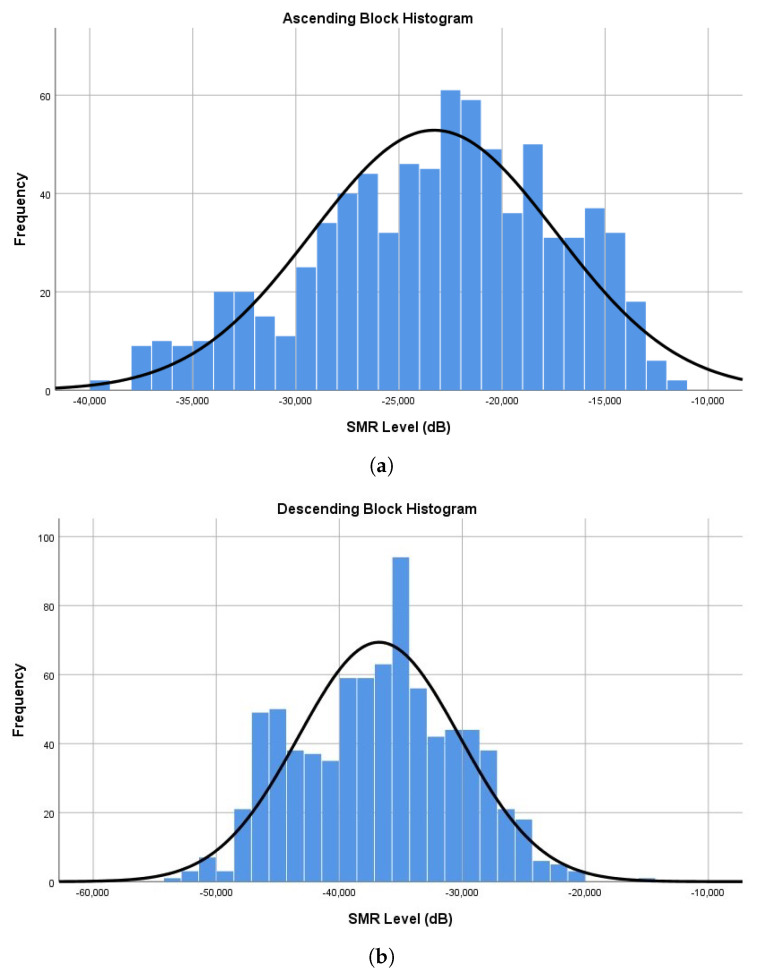
Histogram representation of the obtained SMR levels (in dB). (**a**) Ascending block. (**b**) Descending block.

**Figure 3 ijerph-20-00123-f003:**
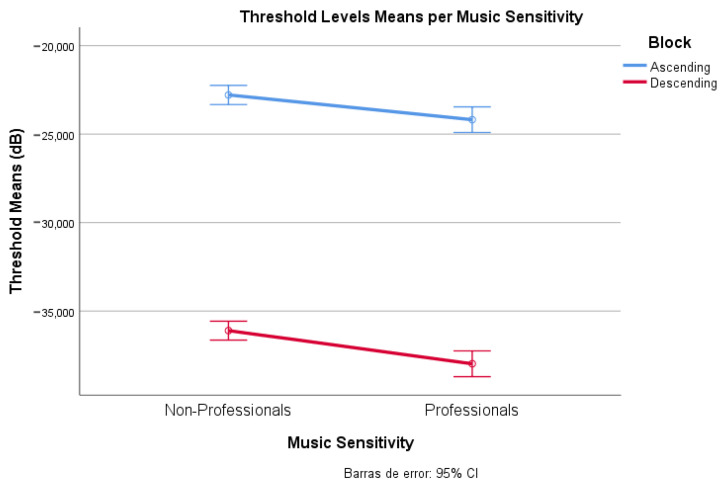
Average threshold levels difference of music sensitivity.

**Figure 4 ijerph-20-00123-f004:**
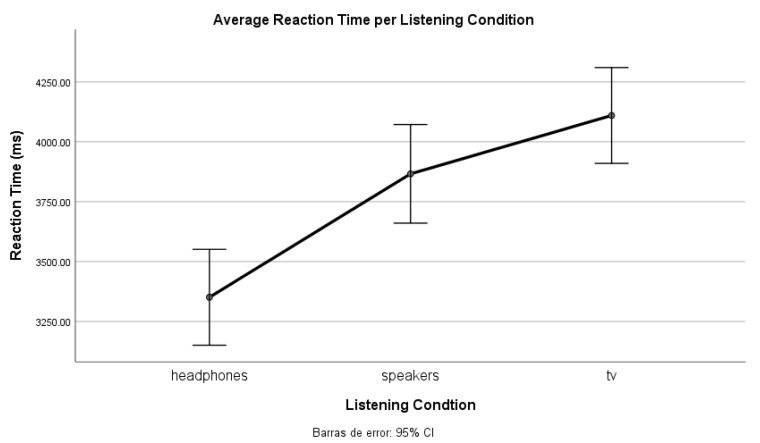
Average reaction time per listening condition.

**Figure 5 ijerph-20-00123-f005:**
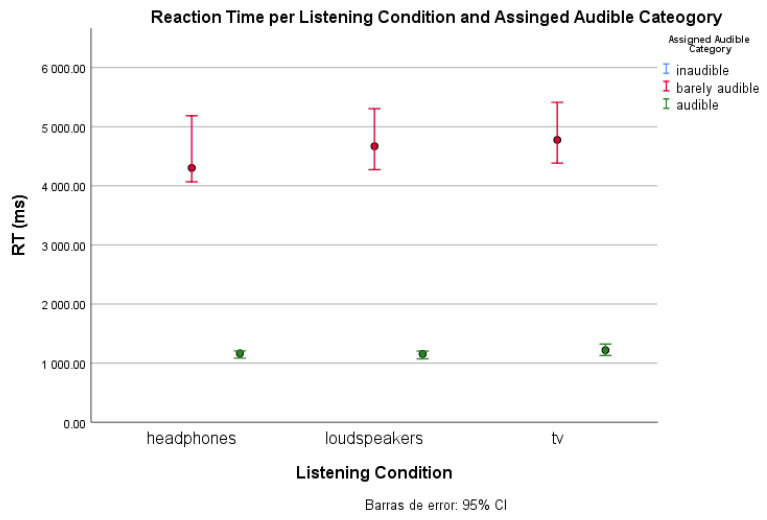
Median reaction time per listening condition and audibility category. There is no representation of the inaudible category as the reaction time is considered null.

**Table 1 ijerph-20-00123-t001:** Average and media volumes set by the participants.

Condition	Headphones	Loudspeakers	TV
Device average volume	9.8	3.5	22.3
Device median volume	6	30	23
Median in dBA	43	48	57

**Table 2 ijerph-20-00123-t002:** Mean threshold values per type of video and experiment block.

Type of Content	Ascending	Descending
Advertisement	−25.00053	−37.83150
Cooking show	−26.22300	−35.11954
Documentary	−28.55084	−33.22362
News	−21.12964	−36.85422
Reality show	n.a.	−37.07983
Sports	n.a.	−34.74023
Talk show	−21.03549	−38.29974
TV contest	−24.26186	−35.86748
Culture	n.a.	−42.69381
Weather	−21.56692	n.a.

## Data Availability

The reported study materials and data can be found in https://github.com/roserbatlle/loudsense (accessed on 31 October 2022).

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
