# Peer review of "Towards a Characterization of Background Music Audibility in Broadcasted TV"

_ijerph, 2022, doi:10.3390/ijerph20010123_

Round 1
Reviewer 1 Report
Article needs improvement, both in terms of general layout of the article (where do the authors want to get to?) and in terms of clarity and soundness of the descriptions. Minor to moderate improvements to the English language required.
Main comments and areas to be improved:
Affiliation: what is BMAT. Should be spelled out, and information must be given here. Should appear in Abbreviations used as well as in section of "Conflicts of Interest" - authors must be open and clear here.
Abstract: the very first phrase talks about "auditory masking" - is this actually true, is the article only about auditory masking, or should visual "masking" or distraction (or as later described "inattention deafness") not be taken into account as well. This non-clarity of whether "auditory" or "audio-visual" goes through the entire article.
Chapter 1.1: "The industrial problem" or "Industry problem"?
Line 42: relative loudness rather than just loudness???
Reference 11: the text on and about reference 11 is not acceptable, neither scientifically nor generally in a scientific paper to be published. 152nd AES Convention was not celebrated in May 2022, but "held". In reference, all other references are complete, often with direct links - but reference 11 is not even complete (no mention of AES convention in references section). If direct link is available for full paper version, should be included. Phrase starting line 139 "Furthermore, the ground truth in mentioned work..." to be fully deleted or to be fully rewritten.
Conclusion (and end of Discussion chapter) can be made both more concise and clearer
Author Response
Dear reviewer, we kindly thank you for taking the time and dedication to read our manuscript. Moreover, we thank you for your comments because they will result in a better-quality printed article. In the following responses, we hope to have resolved and clarified all the flaws and requests indicated. A thorough review of the article has been made and we edited the text to improve clarity and intelligibility.
Point 1: Affiliation: what is BMAT. Should be spelled out, and information must be given here. Should appear in Abbreviations used as well as in section of "Conflicts of Interest" - authors must be open and clear here.
Response 1: BMAT means Barcelona Music Audio Technologies. Otherwise known as BMAT Music Innovators (BMAT Licensing, S. L.), is a company involved in the monitoring of music usage. It has been added in the Abbreviation section. Moreover, we have included more details in the Conflicts of Interest section regarding affiliations and received fundings.
Point 2: Abstract: the very first phrase talks about "auditory masking" - is this actually true, is the article only about auditory masking, or should visual "masking" or distraction (or as later described "inattention deafness") not be taken into account as well. This non-clarity of whether "auditory" or "audio-visual" goes through the entire article.
Response 2: The abstract has been reviewed and adjusted according to the comment.
Point 3: Chapter 1.1: "The industrial problem" or "Industry problem"?
Response 3: The chapter title has been changed to “Problem identification”.
Point 4: Line 42: relative loudness rather than just loudness???
Response 4: The issue has been changed according to the comment.
Point 5: Reference 11: the text on and about reference 11 is not acceptable, neither scientifically nor generally in a scientific paper to be published. 152nd AES Convention was not celebrated in May 2022, but "held". In reference, all other references are complete, often with direct links - but reference 11 is not even complete (no mention of AES convention in references section). If direct link is available for full paper version, should be included. Phrase starting line 139 "Furthermore, the ground truth in mentioned work..." to be fully deleted or to be fully rewritten.
Response 5: The text on and about reference 11 has been modified according to the comment in understanding that our claims have not been properly grounded. Reference 11 has been adjusted in the Bibliography section. The phrase "Furthermore, the ground truth in mentioned work..." has been deleted.
Point 6: Conclusion (and end of Discussion chapter) can be made both more concise and clearer.
Response 6: Conclusion and Discussion chapters have been made more concise and clearer.
Reviewer 2 Report
This is an issue that lies behind the paper, namely, is income for us the central feature of the "common good"? Is our culture so saturated with audible, barely audible, and inaudible background manipulation that we are unconsciously being turned into robots of buying machines that compromise and undermine our relational abilities to live together in love, justice, and peace, and instead turn us against each other? Are groups with counter-cultural messages, like the church, sucked into this same manipulative sales game?
p. 10, 5 lines from the bottom, "an" should probably be "a"
p. 13, last paragraph, second last line, "reclaims" rather than "reclaim"
More serious, "NOVA" and "HSD" are omitted from the list of Abbreviations on p. 15.
This is complicated, but the paper probably does what it says it does.
Author Response
Dear reviewer, we kindly thank you for taking the time and dedication to read our manuscript. Moreover, we thank you for your comments because they will result in a better-quality printed article. In the following responses, we hope to have resolved and clarified all the flaws and requests indicated. A general overview of the article has been made and we edited the text to improve clarity and intelligibility.
Point 1: This is an issue that lies behind the paper, namely, is income for us the central feature of the "common good"? Is our culture so saturated with audible, barely audible, and inaudible background manipulation that we are unconsciously being turned into robots of buying machines that compromise and undermine our relational abilities to live together in love, justice, and peace, and instead turn us against each other? Are groups with counter-cultural messages, like the church, sucked into this same manipulative sales game?
Response 1: Although we personally share the concerns expressed by the reviewer, especially those about the unjustified saturation of auditory stimuli in our environment, for us - the joined team between the Music Technology Group at Universitat Pompeu Fabra and BMAT Music Innovators, a company involved in the monitoring of music usage - it is important to define correctly the different audibility level categories (audible, barely audible and inaudible) to guarantee that the stipulated distribution rights according to the regulation by WIPO is achieved. Consequently, our paper adopts an empirical perspective in order to address a practical problem that is indeed framed into a larger socio-economical context that can and should be considered with a critical perspective. But providing such a critical view is outside the scope of our research as we do not have the goals, methods and techniques required to do that in a proper way. This would imply going outside our own expertise and invade the competences of other scholars that may approach this perspective better.
Point 2: p. 10, 5 lines from the bottom, "an" should probably be "a"
Response 2: Corrected.
Point 3: p. 13, last paragraph, second last line, "reclaims" rather than "reclaim"
Response 3: Sentence has been changed to “draws attention to”.
Point 4: More serious, "ANOVA" and "HSD" are omitted from the list of Abbreviations on p. 15.
Response 4: Corrected. All abbreviations have been added to the Abbreviations section.